# The Methyltransferase HemK Regulates the Virulence and Nutrient Utilization of the Phytopathogenic Bacterium *Xanthomonas citri* Subsp. *citri*

**DOI:** 10.3390/ijms23073931

**Published:** 2022-04-01

**Authors:** Yu Shi, Xiaobei Yang, Xiaoxin Ye, Jiaying Feng, Tianfang Cheng, Xiaofan Zhou, Ding Xiang Liu, Linghui Xu, Junxia Wang

**Affiliations:** 1Integrative Microbiology Research Centre, College of Plant Protection, South China Agricultural University, Guangzhou 510642, China; yushi201503@163.com (Y.S.); belinda0213@163.com (X.Y.); yxx13413646894@163.com (X.Y.); fengjiaying0330@163.com (J.F.); chengtianfang@stu.scau.edu.cn (T.C.); xiaofan_zhou@scau.edu.cn (X.Z.); dxliu0001@scau.edu.cn (D.X.L.); 2Guangdong Province Key Laboratory of Microbial Signals and Disease Control, South China Agricultural University, Guangzhou 510642, China

**Keywords:** *Xanthomonas citri* subsp. *citri*, HemK, RNA-seq, motility, exoenzyme, biofilm, virulence, stress tolerance

## Abstract

Citrus canker, caused by the bacterium *Xanthomonas citri* subsp. *citri* (*Xcc*), seriously affects fruit quality and yield, leading to significant economic losses around the world. Understanding the mechanism of *Xcc* virulence is important for the effective control of *Xcc* infection. In this report, we investigate the role of a protein named HemK in the regulation of the virulence traits of *X**cc*. The *hemK* gene was deleted in the *Xcc* jx-6 background, and the Δ*hemK* mutant phenotypically displayed significantly decreased motility, biofilm formation, extracellular enzymes, and polysaccharides production, as well as increased sensitivity to oxidative stress and high temperatures. In accordance with the role of HemK in the regulation of a variety of virulence-associated phenotypes, the deletion of *hemK* resulted in reduced virulence on citrus plants as well as a compromised hypersensitive response on a non-host plant, *Nicotiana benthamiana*. These results indicated that HemK is required for the virulence of *Xcc*. To characterize the regulatory effect of *hemK* deletion on gene expression, RNA sequencing analysis was conducted using the wild-type *Xcc* jx-6 strain and its isogenic Δ*hemK* mutant strain, grown in XVM2 medium. Comparative transcriptome analysis of these two strains revealed that *hemK* deletion specifically changed the expression of several virulence-related genes associated with the bacterial secretion system, chemotaxis, and quorum sensing, and the expression of various genes related to nutrient utilization including amino acid metabolism, carbohydrate metabolism, and energy metabolism. In conclusion, our results indicate that HemK plays an essential role in virulence, the regulation of virulence factor synthesis, and the nutrient utilization of *Xcc*.

## 1. Introduction

*Xanthomonas citri* subsp. *citri (Xcc*) causes bacterial citrus canker, which is one of the most studied phytopathogens in the *Xanthomonas* genus. *Xcc* can attach to the surface of citrus leaves or fruits and invade host tissues via the plant’s natural openings or wounds when in a warm and humid climate [1,2]. The establishment of *Xcc* infection inside host cells causes necrotic lesions on the leaves, stems, and fruits, leading to defoliation, twig dieback, and blemished fruit and, in serious cases, causes premature fruit drop and eventually the death of infected plants. Citrus canker is considered to be one of the most serious citrus quarantine diseases worldwide and remains a serious challenge for all citrus-producing countries.

*Xcc* has been used as a model organism to study pathogenesis and find new solutions for the disease control of *Xanthomonas* [3,4]. In the past few decades, our understanding of the molecular interaction between citrus and *Xcc* has been advanced by the elucidation of the genome sequences of the *Xanthomonas* genus [5,6]. The complete genome of *Xcc* strain 306 contains 2710 genes that are assigned to functions, of which approximately 6% are involved in pathogenicity and host adaptation [7]. *Xcc* utilizes numerous virulence factors for survival on plant surfaces and fitness in hosts for nutrition and pathogenicity [8]. Such factors include lipopolysaccharides (LPS) [9,10], extracellular polysaccharides (EPS) [11], motility, biofilm formation [12,13], and multiple effectors from its secretion systems [14]. Among them, the type III secretion system (T3SS) enables the translocation of several effector proteins from the bacteria to host cells that affect host signaling and metabolism, leading to hypersensitive reactions and pathogenicity responses in the host cell [15,16]. The type II secretion system (T2SS) enables the secretion of numerous hydrolases, such as cellulases, esterases, and proteinase, as virulence factors for pathogenesis [17,18]. Both T2SS and T3SS are required for the full virulence of *Xcc* during the early development of citrus canker symptoms [19,20].

HemK/PrmC-class methyltransferases are conserved from bacteria [21] and yeast [22,23] to humans [24], and are responsible for the methylation of glutamine at the N5 position in the conserved GGQ motif of release factors. A few reports have been published on the important biological functions of HemK in plants [25,26,27], yeast [22,23], and bacteria [28,29,30]. In *Arabidopsis*, NRF1, a HemK-class glutamine-methyltransferase involved in the termination of translation, is essential for cellular iron homeostasis and the plant’s normal growth [25]. In *Saccharomyces cerevisiae*, there are two protein methyltransferases, Mtq1p (YNL063w) and Mtq2p (YDR140w). The depletion of Mtq1p leads to moderate growth defects on non-fermentable carbon sources and increases the readthrough of a stop codon present in Cox2 mRNA [23]. In contrast, the deletion of the *mtq2* gene displays growth restriction and sensitivity to low temperature, high salt, and calcium concentrations on common yeast medium YPD, as well as sensitivity to translation fidelity antibiotics such as paromomycin and geneticin [22]. In *Pseudomonas aeruginosa* PA14, loss of PrmC activity abolishes anaerobic growth and results in reduced pathogenicity in the infection model *Galleria mellonella* and production of several virulence factors, such as pyocyanin, rhamnolipids and the type III-secreted toxin ExoT [28]. In the *Escherichia coli* K12 strain, *hemK* deletion leads to increased stop codon readthrough, induction of the oxidative stress response, and severely retarded growth [21,30]. In *Yersinia pseudotuberculosis*, the *vagH* (homology to HemK of *E. coli*) mutant exhibits a virulence phenotype similar to that of a T3SS-negative mutant, indicating a close link between VagH and T3SS in *Y. pseudotuberculosis* [29]. The importance of HemK/PrmC class methyltransferase, which has been reported to mediate diverse cellular processes involved in pathogenesis, development, and environmental adaption in some other organisms, prompted us to investigate its specific role in the *Xcc* strain, which is encoded by *XAC0908* and has an unknown biological function.

In this study, we dissected the functional involvement of HemK, a glutamine-methyltransferase from *Xcc*, in virulence through testing a series of phenotypes, and its regulation of gene expression through RNA sequencing analysis. The mutant Δ*hemK* displayed several virulence-related phenotypes, manifested by reduced motility, extracellular enzyme and polysaccharide production, biofilm formation, and pathogenicity. This discovery provides new information on the pathogenicity of this important plant pathogen.

## 2. Results

### 2.1. HemK Influences the Cell Motility, Biofilm Formation, Extracellular Polysaccharide, and Enzyme Production in Xcc jx-6

To elucidate whether HemK plays a role in the cellular process associated with pathogenesis in *Xcc*, we conducted a series of bacterial phenotypic tests to examine the influence of *hemK* deletion on virulence-associated traits, including cell motility, biofilm formation, extracellular polysaccharides, and enzyme production. For those experiments, we engineered an in-frame deletion mutant Δ*hemK* in wild type strain *Xcc* jx-6 background and constructed its complementary strain C-∆*hemK* by introducing a recombinant plasmid pBBR1-*hemK* into Δ*hemK*, which expressed full-length HemK under an arabinose inducible promoter. A phenotypic comparison was performed between wild-type strain *Xcc* jx-6, the mutant strain Δ*hem**K*, and the complementary strain C-∆*hem**K*, to determine the effect of *hemK* deletion on virulence-related traits.

The growth characteristics of those three strains, grown in liquid yeast extract broth (YEB) medium, were first tested with the Bioscreen C automated microbiology growth curve analysis system. The growth curve of Δ*hemK* did not show obvious differences compared with that of wild-type and C-∆*hemK* (Figure 1A). Rich medium YEB was selected for subsequent bacterial phenotyping test.

Then, we examined the swimming motility (tested on 0.3% agar plates) and swarming motility (tested on 0.6% agar plates) of those three strains to probe the roles played by HemK in bacterial motility. Mutant ∆*hemK* showed considerably reduced swimming and swarming motility compared to the wild-type strain. In the complementary strain C-∆*hemK*, these reduced motilities can be restored (Figure 1B,C). Furthermore, in the biofilm formation assay, a significant difference was also observed in ∆*hemK* compared with the wild-type strain by quantifying the cells fixed at the air–media interface of glass tubes using crystal violet (CV) staining. The ∆*hemK* mutant produced approximately half the amount of biofilm as the wild-type strain, whereas the complementary strain C-∆*hemK* produced wild-type-level biofilm (Figure 1D). These results indicate that HemK regulates motility and biofilm formation in *Xcc* jx-6.

Extracellular polysaccharides (EPS) are considered to be an important virulence factor involved in biofilm formation during the disease processes in many bacteria [31]. We then quantified EPS production in wild-type, ∆*hemK* and C-∆*hemK* strains, and found that the ∆*hemK* mutant exhibited significantly decreased EPS production by 40% compared with the wild-type strain. However, the complementary strain C-∆*hemK* produced a similar quantity of EPS as the wild-type strain. Furthermore, the reduction in EPS production caused by HemK disruption was also confirmed by the smaller colony sizes of the ∆*hemK* mutant, grown on YEB agar plates, than that of the wild-type strain (Figure 1E). These results indicate that HemK regulates EPS production in the *Xcc* jx-6 strain.

Finally, a comparison of the production of the activity of cellulases, proteases and amylases for the wild-type, the ∆*hemK*, and C-∆*hemK* strains was conducted using radial diffusion assays based on a calculation of the clearance area of the hydrolysis zones in extracellular enzyme activity assays. The results showed a significant reduction in the activity of these three enzymes in ∆*hemK* when compared with wild-type and C-∆*hemK* strains, indicating that HemK is involved in regulating the production of these extracellular enzymes in *Xcc* jx-6 (Figure 1F–H).

### 2.2. HemK Contributes to Bacterial Stress Tolerance of Oxidative Stress and Heat Shock in Xcc jx-6

We then examined the impact of the *hemK* deletion on the bacteria’s tolerance of several environmental stresses, and the results showed that the Δ*hemK* mutation did not significantly affect the response to some stresses, such as ultraviolet (UV) radiation, saline stress, and osmotic challenge (Appendix A). However, the Δ*hemK* mutant becomes much more sensitive to high temperature and hydrogen peroxide (H_2_O_2_)-induced oxidative stress. As shown in Figure 2, the survival level of the wild-type strain did not show obvious differences between the bacterial cells treated with 0.8, 1.6, and 3.2 mM H_2_O_2_ and the untreated controls (the colony appeared at a concentration of from 10^4^ to 10^5^ CFU/mL); however, the ∆*hemK* mutant that was exposed to 3.2 mM H_2_O_2_ (the colony appeared at a concentration of 10^8^ CFU/mL) displayed a significantly decreased cell survival and viability compared with the untreated control cells (the colony appeared at a concentration of 10^5^ CFU/mL) (Figure 2A). The survival of 3.2 mM H_2_O_2_-treated bacterial cells in the YEB liquid medium was further quantitatively detected. The ∆*hemK* mutant showed a 70% lower survival rate than the wild-type strain. Similarly, the survival rate of the high temperature-treated ∆*hemK* mutant on the YEB agar surface or in liquid YEB media was obviously reduced from that of the wild type (Figure 2B). In all the experiments performed above, the levels of stress tolerance to both H_2_O_2_ and the high temperature of C-∆*hemK* were comparable with those of the wild-type strain (Figure 2). These results indicate that the mutation of *hemK* reduces bacterial tolerance to oxidative stress and heat shock in *Xcc*.

### 2.3. Mutation of hemK Impaired Activation of Virulence on Citrus and Hypersensitive Response to Nicotiana benthamiana

The above finding that *hemK* deletion resulted in the reduced production of several virulence factors prompted us to further explore whether HemK influences the virulence of *Xcc*. For this, the pathogenicity of wild-type *Xcc* jx-6, Δ*hemK*, and C-Δ*hemK* strains to the host plant, the Hongjiang sweet orange, was tested as described in the Materials and Methods section. The bacterial cells of these three strains, grown in XVM2 to 10^5^ CFU/mL, were sucked into the needleless syringes and introduced onto the leaves of the sweet orange. Ten days post-inoculation, the *Xcc* jx-6 pathogen-infected citrus leaves showed obvious canker disease symptoms, characterized by water-soaking phenotypes, while the Δ*hemK* mutant produced only very mild disease symptoms on the leaves. The area of water-soaked lesions on leaves inoculated with the Δ*hemK* mutant decreased to about 20% of that on the wild-type-inoculated leaves. Furthermore, the complementary strain C-Δ*hemK* produced similar virulence symptoms as the wild type (Figure 3A,B). These data, together with the findings reported above, suggest that HemK is essential to the virulence of *Xcc*.

In addition, we investigated whether *hemK* deletion has an influence on the ability to induce a hypersensitive response in the non-host plant *N**. benthamiana*. For these experiments, the Δ*hemK* mutant was infiltrated at a cell concentration of 10^5^ CFU/mL into the leaves of *N. benthamiana*. The results showed that the *hemK* mutant elicited a 30% decrease in the hypersensitive response symptoms compared to those of the wild-type strain (Figure 3C,D), suggesting that the HemK trigger compromised the hypersensitive response in its non-host plant *N. benthamiana*.

### 2.4. Transcriptome RNA Sequencing (RNA-Seq) Analysis Reveals Multiple Physiological Processes Regulated by HemK in Xcc

To gain insight into the global regulatory impact of HemK in controlling the virulence of *Xcc* at a transcriptional level, we performed transcriptomic analysis for wild-type *Xcc* jx-6 and its isogenic Δ*hemK* mutant. For this analysis, the *Xcc* strains were grown to OD_600_ ≈ 0.8 in the minimal medium XVM2, which is closer to the nutrition environment of the plant intercellular spaces and induces the expression of a series of virulence-related genes [32].

The cells were then collected for total RNA extraction and library construction. The constructed libraries were sequenced using the Illumina Hiseq 2000 platform. Differentially expressed genes (DEGs) between Δ*hemK* and the *Xcc* jx-6 strain were determined using DESeq software. Comparative RNA-seq data analysis was performed as described in the Materials and Methods section. Finally, of the 4489 annotated genes from the genome of the *Xcc* strain 306, a total of 286 genes were determined as significant DEGs between those two strains. Of these, 166 genes were downregulated and 120 genes were upregulated in ∆*hemK*, with 11 genes being identified as novel genes in this assay (Appendix A).

To classify the above-identified 286 DEGs into biological function groups, the clusters of orthologous genes (COG) enrichment analysis was employed for those genes according to the genome of the *Xcc* strain 306. In the COG analysis, based on the conserved domain alignment, a total of 171 DEGs were successfully annotated and grouped into 20 COG functional categories; all the results for each category are presented in Figure 4. Except for those terms with functions unknown (Figure 4, columns R and S), the three most enriched functional COG terms were related to “carbohydrate transport and metabolism” (19 members, 11.1%), “amino acid transport and metabolism” (17 members, 9.9%), and “lipid transport and metabolism” (16 members, 9.4%). Other enriched terms included “energy production and conversion” (12 members, 7.0%), “replication, recombination and repair” (11 members, 6.4%), “cell wall/membrane/envelope biogenesis” (11 members, 6.4%), “signal transduction mechanisms” (10 members, 5.8%), “inorganic ion transport and metabolism” (9 members, 5.3%), “transcription” (7 members, 4.1%), “secondary metabolites biosynthesis transport and catabolism” (5 members, 2.9%) and “coenzyme transport and metabolism” (5 members, 2.9%). These results indicated that the DEGs significantly changed by *hemK* deletion were dominantly involved in “metabolism” and “signal transduction”.

To further identify the most significant metabolic or signal transduction pathways for all DEGs, a Kyoto Encyclopedia of Genes and Genomes (KEGG) pathway enrichment analysis was performed. By comparison of 286 DEGs with the whole *Xcc* strain 306 genome background, a total of 91 DEGs, including 59 down- and 32 up-regulated genes, were matched into 47 pathways that were gathered into four categories, including “metabolism” (37 members), “cellular processes” (3 members), “environmental information processing” (3 members) and “genetic information processing” (4 members) in the KEGG database (Figure 5, Appendix A). The top 20 enriched KEGG pathways are shown in Figure 5.

Annotation results of 51 DEGs involved in “metabolism”, which represented the dominant category in the KEGG pathway, were sorted into 7 subcategories. Those subcategories were “global and overview maps” (33 DEGs, 6 pathways), “carbohydrate metabolism” (12 DEGs, 11 pathways), “energy metabolism” (4 DEGs, 2 pathways), “nucleotide metabolism” (1 DEG, 1 pathway), “amino acids metabolism” (32 DEGs, 10 pathways), “metabolism of cofactors and vitamins” (9 DEGs, 6 pathways), and “biosynthesis of other secondary metabolites” (1 DEG, 1 pathway) (Appendix A). In addition, as listed in Figure 5, a number of DEGs were found to be associated with bacterial virulence, which were sorted into subcategories, including “bacterial secretion system”, “two-component system”, “quorum sensing”, “flagellar assembly” and “bacterial chemotaxis” (Figure 5, Appendix A).

Together, the results from the COG and KEGG enrichment analyses reveal that HemK mediates multiple physiological processes, including nutrient utilization and the virulence of *Xcc*.

### 2.5. HemK Controls the Expression of T3SS and Its Associated Effectors, as Well as Many Extracellular Enzymes Secreted by T2SS

T3SS is required for the full virulence of *Xcc* by translating the diverse proteins referred to type-III-secreted effectors into plant host cells to influence bacterial pathogenicity. The transcriptome profile data presented above revealed the significantly decreased expression of 43 genes associated with T3SS in the ∆*hemK* mutant. These genes included 24 genes of the whole *hrp*/*hrc* gene cluster coding for the core structural component of T3SS apparatus (*hrpF*, *XAC0395*, *hpaB*, *hrpE*, *hrpD6*, *hrpD5*, *hpaA*, *hrcS*, *hrcR*, *hrcQ*, *hpaP*, *hrcV*, *hrcU*, *hrpB1*, *hrpB2*, *hrcJ*, *hrpB4*, *hrpB5*, *hrcN*, *hrpB7*, *hrcT*, *hrcC*, *hpa1* and *hpa2*), 18 genes encoding for putative T3SS-associated effectors (*hpaF*, *XAC0419*, *avrBs2*, *xopR*, *avrXacE1*, *xopX*, *xopI*, *xopN*/*HopAU1*, *hrpW*, *xopK*, *xopL*, *XAC3230*, *XAC3666*, *xopQ*/*HopQ1*, *xopAU*, *xopAV*, *xopAP*, and *xopS*), and *hrpX*, encoding a transcriptional regulator of virulence (Figure 6A, Appendix A) [15,33].

T2SS secretes numerous hydrolases, such as cellulases, proteases, and xylanases as virulence factors to contribute to canker symptom development in the *Xanthom**onas* genus [12,17,18]. RNA-seq data also revealed a decreased expression of 10 genes in ∆*hemK*, encoding bacterial exoenzymes secreted through T2SS (*virK*, *XAC0346*, *XAC0612*, *XAC0795*, *XAC0933*, *XAC2831*, *XAC2833*, *XAC2853*, *XAC3490* and *XAC3545*) (Figure 6A, Appendix A).

To further confirm whether HemK is required for the regulation of T3SS- and T2SS-associated gene expression, as revealed by the RNA-seq analysis, we conducted quantitative real-time PCR (qRT-PCR) for wild-type *Xcc* jx-6 and ∆*hemK* strains to evaluate the relatively endogenous mRNA level of 11 genes in *Xcc*. These selected genes included *hrpB2*, *hpaA*, *hrcC*, and *hrpE* (T3SS regulators), *avrXacE1*, *avrBs2* and *virK* (virulence protein), *XAC0612* (cellulase), *XAC3545* (protease), *XAC3490* (amylosucrase or alpha-amylase) and *XAC2853* (cysteine protease). As expected, the qRT-PCR results display a significantly decreased mRNA level of these selected 11 genes in the ∆*hemK* mutant compared with the wild type (Figure 6B).

Given the finding that the deletion of *hemK* results in reduced exoenzyme secretion (Figure 1F–H) and impaired activity, to trigger the hypersensitive response in its non-host plant, *N. benthamiana* (Figure 3C,D), it is reasonable to suspect that the HemK-positive regulation of the T3SS- and T2SS-associated genes’ expression at the transcriptional level accounts for the phenotypes that appeared in the ∆*hemK* mutant.

### 2.6. HemK Is Implicated in the Regulation of the Expression of Genes Involved in Diffusible Signal Factor (DSF) Mediated Quorum Sensing of Xcc

Previous transcriptome profile and proteomic analyses revealed that DSF-mediated quorum sensing specifically modulates bacterial adaptation, nutrition uptake and metabolism, stress tolerance, virulence, and signal transduction to favor host infection [34,35]. We compared the transcript profiles of the ∆*hemK* mutant (Appendix A) and that of the previously reported DSF-deficient (Δ*rpfF*) mutant during citrus infection in planta [34], and observed 32 genes being regulated by both HemK and RpfF, of which 31 genes were downregulated in ∆*hemK* (Table 1).

## 3. Discussion

HemK is evolutionarily conserved from prokaryotes to eukaryotes [22,24,30], but only a few orthologous genes have been studied so far. The *hemK*-knockout mutation in mice was lethal [36], while the loss of HemK results in severe growth defects in some bacteria [21] and increased sensitivity to low temperature, salinity, and high calcium concentrations in addition to growth restriction in yeast [22]. In *Arabidopsis*, the methyltransferase NRF1 knockout line displays imbalances in cellular ion levels, severe growth retardation, and demonstrates various developmental defects [25]. However, little is known about the physiological role of HemK in the phytopathogen *Xcc*. In this study, we demonstrated that the Δ*hemK* mutant showed decreased motility, biofilm formation, and the production of cellulases, amylases, and proteases, as well as decreased adaptation to oxidative stress and heat shock. Furthermore, the virulence of ∆*hemK* on host and non-host plants, respectively, was significantly decreased. HemK was further shown by transcriptomic analysis to play a role in the regulation of a variety of genes involved in bacterial virulence, secretion systems, chemotaxis, and metabolism. Our studies indicated that HemK may play an essential role in regulating the virulence and environmental adaptation of *Xcc*.

In this study, RNA-seq analysis showed that the genes regulated by HemK included 32 genes related to amino acid uptake and metabolism, 43 genes related to T3SS and its effectors, 12 genes related to carbohydrate metabolism, 18 genes related to chemotaxis and flagellar assembly, and 32 genes related to the diffusible signal factor (DSF)-mediated quorum sensing of *Xcc*. Phenotypical analyses revealed that the deletion of *hemK* caused reduced swimming and swarming motility, extracellular enzymes, and polysaccharide production in *Xcc*. Several reports have shown that the lack of extracellular enzymes reduced the virulence by affecting the early colonization of the pathogen in *Xanthomonas* [13,17]. The mutation of *Bglc 3* in *Xcc* strain 29-1 lost the ability to produce cellulases, resulting in its weakened pathogenicity on citrus [13]. Bacterial motility and chemotaxis contributed to virulence at the initial stages of *Xcc* infection [37,38]. Based on our findings in this report, we speculate that HemK may regulate virulence by affecting the early colonization step of *Xcc* infection during the development of symptoms in citrus. The T3SS encoded by the *hrp* gene cluster is involved in delivering a number of hypersensitive-response elicitors or pathogenic factors into plant cells and is critical for the successful infection and colonization of *Xcc* in host and non-host tissues [12,39,40,41]. HemK and its homologous proteins have been reported to be related to T3SS in different pathogens. For example, the virulence-associated protein VagH in *Y. pseudotuberculosis* shares high homology with the HemK of *E**. coli*, has a methyltransferase activity similar to HemK, and shows a close link with T3SS [29]. In *P. aeruginosa* PA14, HemK, an S-adenosyl-l-methionine (AdoMet)-dependent methyltransferase of peptide chain release factors, is essential for the expression of virulence factors and the type III-secreted toxin ExoT [28]. In this study, the transcriptomic analysis showed that HemK positively regulated the expression of T3SS genes, including 24 *hrp* genes. *HrpX* is the master regulator of T3SS and its effector genes in *Xanthomonas* [15,33,42]. Our qRT-PCR analysis further confirmed that a number of T3SS effector genes, including *hrpB2*, *hpaA*, *hrcC*, *hrpE*, and virulence genes (*avrXacE1* and *avrBs2*), are down-regulated in ∆*hemK*. It is, therefore, likely that HemK may regulate the pathogenicity of *Xcc* by regulating the expression of *hrp* genes and T3SS effectors. In summary, our research provides new insights into the functional roles of HemK in regulating the pathogenicity, virulence, and environmental tolerance of *Xcc*. Further investigations may help to understand whether the catalytic activity of HemK is critical for bacterial survival under specific stress conditions and whether HemK is a potential target for antibacterial drug development.

## 4. Materials and Methods

### 4.1. Bacterial Strains, Culture Media, and Culture Conditions

The bacterial strains and plasmids used in this study are listed in Appendix A. The *E. coli* strains were grown at 37 °C in either Luria-Bertani (LB) broth (10 g/L tryptone, 5 g/L yeast extract, and 10 g/L NaCl, pH 7.0) or on LB with agar. The optimum growth temperature for the *Xcc* jx-6 strain was 28 °C, and the strains were grown in NYG medium (5 g/L peptone, 3 g/L yeast extract, 20 mL/L glycerol, pH 7.0), YEB medium (10 g/L tryptone, 5 g/L yeast extract, 5 g/L NaCl, 0.5 g/L MgSO_4_·7H_2_O, 5 g/L sucrose, pH 7.0), or XVM2 medium (20 mM NaCl, 10 mM (NH_4_)_2_SO_4_, 5 mM MgSO_4_, 1 mM CaCl_2_, 0.16 mM KH_2_PO_4_, 0.32 mM K_2_HPO_4_, 0.01 mM FeSO_4_, 10 mM fructose, 10 mM sucrose, 0.03% casamino acids (pH 6.7)). NY medium was NYG medium without glycerol [43]. YEB medium was used for exoenzyme production assays. Antibiotic gentamicin (Sangon Biotech, Shanghai, China) was added at a final concentration of 25 µg·mL^−1^ when required.

### 4.2. Plasmid Construction and Primers Used

DNA fragments encoding the full-length or truncated proteins were amplified with a polymerase chain reaction (PCR) from the genomic DNA of the *Xcc* jx-6 strain. Primers were designed based on the published complete genome sequence of *Xcc* jx-6 (NCBI reference sequence: NZ_CP011827.2) and are listed in Appendix A. Primer synthesis and sequencing services were performed by Beijing Qingke Biotechnology (Beijing, China).

### 4.3. Construction of the ∆hemK Mutant and Its Complementary Strain in Xcc jx-6

The deletion mutant ∆*hemK* was created from the wild-type *Xcc* jx-6 strain by homologous-recombination-based procedures, as described previously by using the knockout plasmid pK18-Δ*hemK*. This plasmid was constructed by cloning the fused upstream- and downstream-homologous fragments of the HemK region into the suicide vector pK18mobSacB-carrying Gm resistance gene [44]. The DNA fragments of up and down homologous arms of *hemK* were amplified by PCR, with primer pairs of hemK-F1/hemK-R2 and hemK-F3/hemK-R4, respectively. The obtained two fragments were fused together by PCR, using the primer pair of hemK-F1/hemK-R4. The fused DNA fragment was digested with HindIII and BamHI and then ligated into the corresponding sites of the vector pK18mobSacB. Then, the obtained knockout vector pK18-Δ*hemK* was transformed into wild-type strain *Xcc* jx-6 by electroporation [45], and positive transformants were selected on LB agar medium, supplemented with gentamicin (25 µg/mL). Colonies resulting from the first crossover events were streaked onto LB agar plates supplemented with 10% sucrose, and sucrose-sensitive colonies were selected as positive colonies. Colonies with *hemK* deleted were further confirmed by sequencing the DNA fragment amplified with the primer pair of hemK-5F and hemK-6R.

To construct the complementary plasmid pBBR-hemK, an 861-bp DNA fragment that covered the entire coding region of HemK protein was cloned into the vector pBBR1MCS-5 [46]. The DNA fragment was amplified by PCR using genomic DNA from strain *Xcc* jx-6 as a template, with the primer pair of hemK-F/hemK-R; it was then digested with SmaI and XbaI and ligated into the corresponding sites of the plasmid pBBR1MCS-5. The ligation mixture was transformed into *E. coli* DH5α-competent cells and positive colonies were selected on LB agar with gentamicin (25 µg/mL). Following PCR-based verification using primers M13/pBAD, the recombinant plasmid was transformed into ∆*hemK* by electroporation to produce the complementary strain that was designated as C-∆*hemK*.

### 4.4. Bacterial Growth Curve

We analyzed the growth of the wild type, ∆*hemK*, and C-∆*hemK* using the Bioscreen C automated microbiology growth curve analysis system (Thermo Lab Systems, Helsinki, Finland) by adapting the method described by Medina et al. [47]. Briefly, a single colony of each strain was grown overnight at 28 °C in LB broth with an antibiotic (25 µg/mL gentamicin). The overnight cultures were adjusted to the same concentration by measuring the OD_600_ values and were then diluted 100-fold in fresh YEB medium. An aliquot of 200 µL of this diluted solution was added into the wells of a 100-well microplate, with the fresh YEB used as a negative control. The OD at 600 nm was measured every 4 h at 28 °C with shaking. Four biological replicates were performed for each strain.

### 4.5. Bacterial Motility Assays

Motility assay was examined as described previously [43,48]. Briefly, bacterial strains were all grown in LB liquid medium at 28 °C with shaking for about 18 h and diluted to an optical density at OD_600_ of 0.8. For swimming motility, 2 µL aliquots of bacterial cells were stabbed into the medium in the center of 0.25% Bacto-agar plates containing 0.03% peptone and 0.03% yeast extract with gentamicin and were inoculated for 60h at 28 °C; for swarming motility, 2 µL aliquots of bacteria cells were spotted onto the surface of the center of 0.6% Bacto-agar plates containing NY medium, supplemented with 2% glucose with gentamicin. Following incubation at 28 °C for 60 h, all the plates were photographed. The motility was quantified by measuring the area of the growth circle around the inoculation site with ImageJ. Each assay was repeated at least three times.

### 4.6. Bacterial Biofilm Formation Assay

The biofilm formation was tested by measuring the ability of bacterial cells to adhere to the glass tubes as described previously with slight modification [11,49]. Bacterial strains were grown in YEB medium at 28 °C with shaking at 200 rpm. Then, 1 mL of bacterial suspensions (OD_600_ = 1.0) were inoculated into a glass tube containing 1 mL fresh YEB medium supplemented with 0.05% L-arabinose and 25 µg/mL gentamicin, then left to stand for 72 h at 28 °C. The culture medium was discarded; the attached bacterial cells were stained with 0.1% (*w*/*v*) crystal violet for 45 min, then washed 3 times with distilled water and dried at 60 °C. The crystal violet remaining on the tube wall was dissolved in 33% (*v*/*v*) acetic acid, and the absorbance was measured at 590 nm with a microplate reader (BioTek, Winooski, VT, USA). Three repeated quantitative measurements were performed.

### 4.7. Extracellular Enzymes Activity Assay

Extracellular enzyme activity assays were examined, as described by [50] with some modifications. Briefly, *Xcc* strains were grown in YEB media at 28 °C for about 18 h, and an aliquot of 1.5 µL bacterial culture (adjusted to an OD_600_ of 0.8) was spotted onto the surface in the center of YEB agar plates supplemented with 1% (*w*/*v*) potato starch (for amylase detection), or 1% (*w*/*v*) sodium carboxymethyl ethyl cellulose (for cellulase detection), together with 0.05% (*w*/*v*) L-arabinose and gentamicin. All plates were incubated for 2 days at 28 °C. For protease detection, the bacterial supernatant was collected by centrifuge at 13,000 rpm for 5 min, then a 20 µL aliquot of the supernatant was added to the wells of YEB plates containing 1% (*w*/*v*) milk [51], punched with a hole punch, blown dry, and incubated for 48 h at 28 °C. Three replicates were used for each assay.

### 4.8. Extracellular Polysaccharides Production Assays

Extracellular polysaccharides (EPS) in bacterial culture supernatants were determined quantitatively, as described previously [52]. Bacterial strains were grown in liquid YEB medium with gentamicin overnight until the OD_600_ reached 2.5. The supernatant of the cell cultures was obtained after removing the cell pellets by centrifugation (5000 rpm for 30 min). Two volumes of ethanol were added to the supernatant and the mixtures were incubated at −20 °C overnight. The precipitated EPS was obtained by centrifugation (5000 rpm for 20 min) and the dry weights of EPS were recorded after drying overnight at 65 °C.

EPS production was established by the plate assay; all strains were grown in YEB medium until the OD_600_ reached 0.8. An aliquot of 2 µL bacterial culture was spotted onto the center of a YEB agar plate supplemented with 25 µg/mL gentamicin and 0.05% (*m*/*v*) L-arabinose. Following growth at 28 °C for 2 days, the YEB agar plates were photographed. These experiments were independently repeated at least three times.

### 4.9. Stress Tolerance Assay

The stress tolerance assay was performed as previously described, with modifications [53]. Environmental stresses, including ultraviolet (UV) radiation, saline stress, and osmotic challenge, heat shock, and oxidative stress challenges were measured. The bacterial strains of the mutant ∆*hemK*, the wild type, and the C-∆*hemK* strain were cultured in an LB medium containing gentamicin to the early exponential stage (OD_600_ of 0.7) and were collected for stress treatment. For oxidative stress treatment, the bacterial cultures were exposed to three different H_2_O_2_ concentrations, namely, 0.8, 1.6, and 3.2 mM for 30 min. For heat shock treatment, the bacterial cultures were heated at 50 °C for periods of 0, 80, 160, and 320 s. For survival growth tests of stress-treated cells, the treated bacterial cultures (adjusted to a concentration of 10^8^ CFU/mL) were serially diluted by a factor of ten to 10^3^ CFU/mL, and a 2 µL aliquot of each ten-fold dilution sample was spotted onto the surface of the YEB agar plates and incubated at 28 °C for 48 h. By a comparison of the growth of colonies before and after treatment, the levels of stress tolerance were determined.

In the quantitative analysis experiment, the stress-treated bacterial culture was inoculated into fresh YEB medium at a 1:50 ratio, and the diluted solution was cultured via shaking for 28 h, then the optical density at 600 nm (OD_600_) was measured. The relative survival rate was defined as the percentage of viable cell counts from the bacterial culture with stress treatment (T1, value of OD_600_) compared with those from the non-treated culture (T0, value of OD_600_). Each treatment was repeated three times, with three replicates for each strain.

### 4.10. Virulence and Hypersensitive Response Assays

Pathogenicity assays were conducted as described previously, with little modification [32,54]. An assay was performed using the immature leaves of the sweet orange (*Citrus sinensis*) grown outdoors at approximately 18 °C to 28 °C. The wild-type *Xcc* jx-6 strain, mutant strain ∆*hemK*, and complementary strain C-∆*hemK* were cultured in XVM2 medium at 28 °C with shaking until the value was 0.4 of OD_600_. First, 1 mL of each cell culture was centrifuged at 10,000 rpm and resuspended in 1 mL of sterile water. Then, 10 µL of each bacterial solution was infiltered into the leaves with a needleless syringe. Sterile water was infiltered in the same way as the negative control. The disease symptoms of the inoculated sweet orange leaves were photographed 10 days after inoculation. The hypersensitive response (HR) assay on *N.benthamiana* was also conducted for these three strains. The tobacco plants were grown in a quarantine greenhouse facility (parameters: light for 16 h and darkness for 8 h at 28 °C). The cells (approximately 10^5^ CFU/mL) were then pelleted by centrifugation and resuspended (1:1) in sterile water, and the disease symptoms of *N. benthamiana* were photographed at 3 days post-inoculation. Experiments were repeated at least three times with similar results.

### 4.11. High-Throughput RNA Sequencing (RNA-Seq) and Data Analysis

A single bacterial colony of the wild-type strain *Xcc* jx-6 and the ∆*hemK* mutant was cultured in 10 mL LB broth at 28 °C with shaking for 24 h, along with three biological repeats performed for each strain. Overnight cultures were adjusted to the same OD_600_, and were then diluted (1:100) in XVM2 media and incubated at 28 °C. Until OD_600_ = 0.8, the cells were collected by centrifugation at 12,000 rpm and immediately frozen with liquid nitrogen for total RNA extraction. The total RNA was extracted using the Eastep Super Total RNA extraction kit (Shanghai Promega Biological Products Ltd, Shanghai, China). The quantity and quality of the total RNA were determined using a NanoDrop (NanoDrop 2000 Technologies, Wilmington, NC, USA), agarose gel electrophoresis, and an Agilent 2100 bioanalyzer (Agilent Technologies, Waldbronn, Germany).

Transcriptome library construction and sequencing were performed using Novogene (Beijing, China). Clean read data were obtained following sequencing and data filtering, and the resulting reads were mapped to the *Xcc* 306 reference genome. Gene expression, as indicated by the expected number of fragments per kilobase of the transcript sequence, per million base pairs sequenced (FPKM), was calculated using HTSeq. DEGs between the strains ∆*hemK* and *Xcc* jx-6 were determined using DESeq software, based on *q*-values of <0.05 and a minimum |log_2_ (Fold Change)| > 1, and were exported as a tabular file (Appendix A). The obtained DEGs between those two strains were then classified and enriched, based on the Clusters Orthologous Groups database (COG, https://www.ncbi.nlm.nih.gov/COG/, accessed on 29 December 2021) and KEGG pathway analyses.

### 4.12. Quantitative Real-Time PCR (qRT-PCR) Assay

To verify the RNA-seq results, a qRT-PCR assay was performed using the total RNA extracted from strains grown under the same growth conditions as the RNA-seq. RNA samples were reverse-transcribed using a HiScript II QRT SuperMix for qPCR (Vazyme Biotech, Nanjing, China). The cDNA was subjected to a two-step qRT-PCR assay, using a ChamQ Universal SYBR qPCR Master Mix (Vazyme Biotech) on a QuantStudio^TM^ 6 Flex System. The 16 s rRNA gene was used as an endogenous control. The relative fold change in the target genes’ expression was calculated using the formula 2-ΔΔCT [55]. The primer sequences used in the qRT-PCR assay are listed in Appendix A.

## Figures and Tables

**Figure 1 ijms-23-03931-f001:**
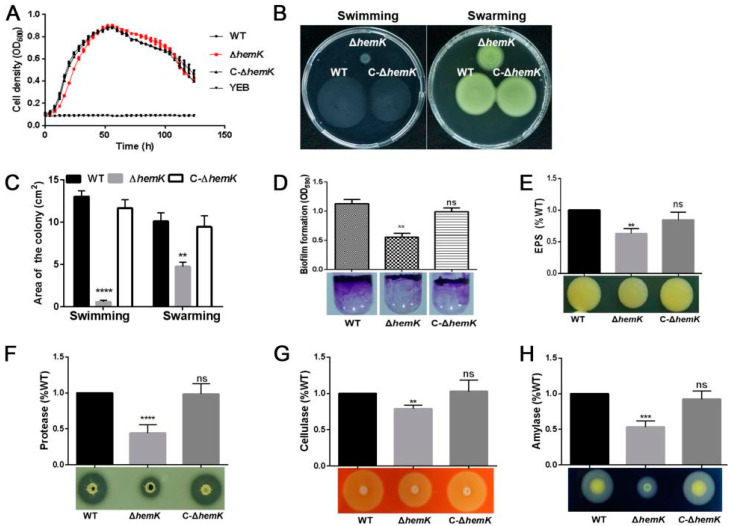
HemK influences the production of cell motility, biofilm formation, extracellular polysaccharides, and enzymes in *Xcc* jx-6. (**A**) The growth rates (OD_600_ values) of the wild-type (WT) strain *Xcc* jx-6, Δ*hemK*, and C-∆*hemK* on YEB at 28 °C were measured at 4 h intervals. (**B**) Swimming and swarming motility for the WT strain, Δ*hemK*, and C-∆*hemK* were detected on a 0.28% agar swimming plate and 0.6% agar swarming plate, respectively. A 2 µL aliquot of the bacterial suspension was inoculated onto the swimming plate or swarming plate and incubated for 60 h at 28 °C to observe the bacterial motility. (**C**) The level of motility was determined by measuring the area of the colony with ImageJ. (**D**) The biofilm formation of the WT strain, ∆*hemK*, and C-∆*hemK* in glass tubes was detected by crystal violet staining and quantified by measuring the optical density at 590 nm, after dissolution in 33% acetic acid. (**E**) The production of extracellular polysaccharides (EPS) of the WT strain, ∆*hemK*, and C-∆*hemK* was assessed on 2% glucose NYGA, and the production was measured with ethanol precipitation. (**F**–**H**) A 20 µL (for proteases) or 2 µL (for cellulases and amylases) aliquot of bacterial supernatant was added to the exoenzymes’ test plates and incubated at 28 °C for 48 h. The production of hydrolysis circles by cellulases, amylases, and proteases was measured on plates containing 1% (*m*/*v*) skimmed milk (**F**), 1% (*m*/*v*) sodium carboxymethyl ethyl cellulose (**G**), and 1% (*m*/*v*) potato starch (**H**), respectively. All experiments were repeated three times, with three repetitions for each strain. Only one representative result is presented. A significant difference between the WT and ∆*hemK* was demonstrated with the respective treatments: **** *p* < 0.0001, *** *p* < 0.001, ** *p* < 0.01, ns: no significance (Student’s *t*-test).

**Figure 2 ijms-23-03931-f002:**
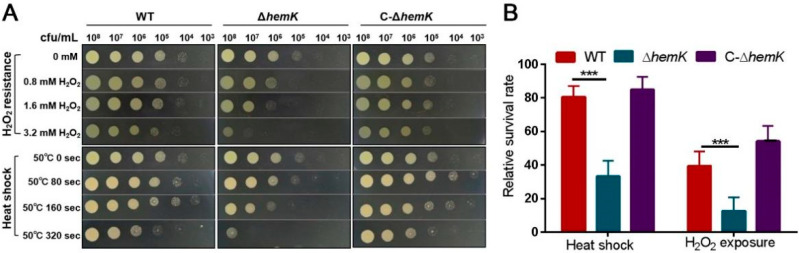
HemK contributes to stress tolerance. (**A**) The tolerance of wild-type *Xcc* jx-6 strain, ∆*hemK*, and C-∆*hemK* was performed under H_2_O_2_-induced oxidative stress and heat shock. (**B**) Bacterial cell viability was estimated by measuring the value of OD_600_ on the YEB medium before (T0) and after (T1) treatment. The survival rate was calculated as the ratio of the cell count at T1 to that at T0. Each test was repeated five times, and these five times had similar results. The data shown are the means and standard errors of five replicates. The significant difference between the WT and ∆*hemK* with respective treatments: *** *p* < 0.001.

**Figure 3 ijms-23-03931-f003:**
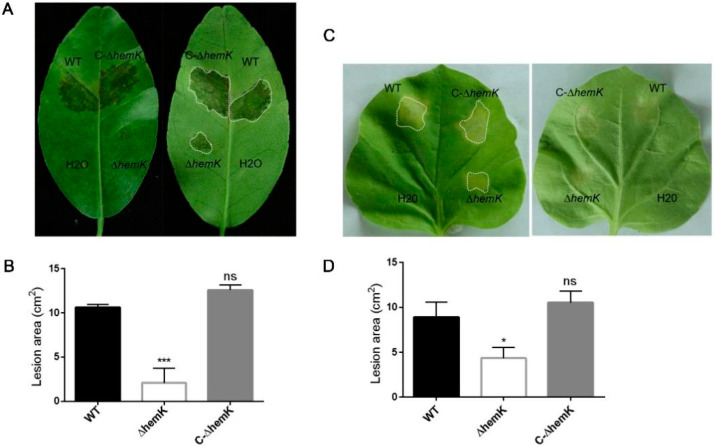
HemK regulates the virulence and hypersensitive response of plants. (**A**) Pathogenicity assay for wild-type, ∆*hemK*, and C-∆*hemK* was performed on citrus leaves. The bacterial suspensions (approximately 10^5^ CFU/mL in XVM2) were inoculated into the young leaves of sweet orange by pressure infiltration with a needleless syringe. Sterile H_2_O was used as a negative control. A representative leaf from three replicates was photographed 10 days post-inoculation. (**B**) The area of the lesions was determined by ImageJ. The means and standard errors of three replicates from one representative result are shown. The significant difference between the WT and ∆*hemK* with respective treatments: *** *p* < 0.001 (Student’s *t*-test). (**C**) The symptoms of hypersensitive response were photographed 3 days post-inoculation on the leaf surface of *N. benthamiana*. (**D**) The area of the infection spot was measured using ImageJ. Sterile H_2_O was used as a negative control. The means and standard errors of three replicates from one representative result are shown. The significant difference between the WT and ∆*hemK*, with respective treatments: * *p* < 0.05, ns: no significance (Student’s *t*-test).

**Figure 4 ijms-23-03931-f004:**
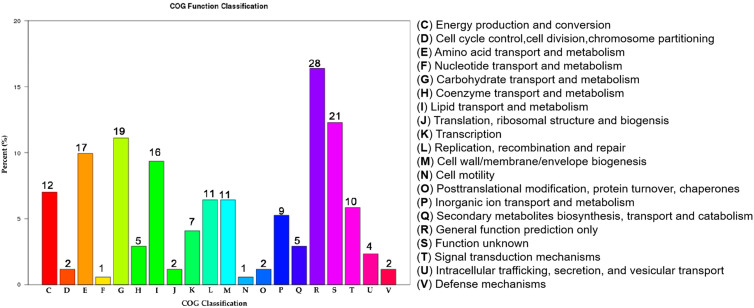
HemK regulates multiple clusters of orthologous genes (COG) functional categories. The *x*-axis represents the functional classification of each COG category. The *y*-axis represents the relative abundance (%) of DEGs in each COG category. The number of genes (presented above each category) in different functional classes reflects the metabolic and physiological bias in a certain period or environment.

**Figure 5 ijms-23-03931-f005:**
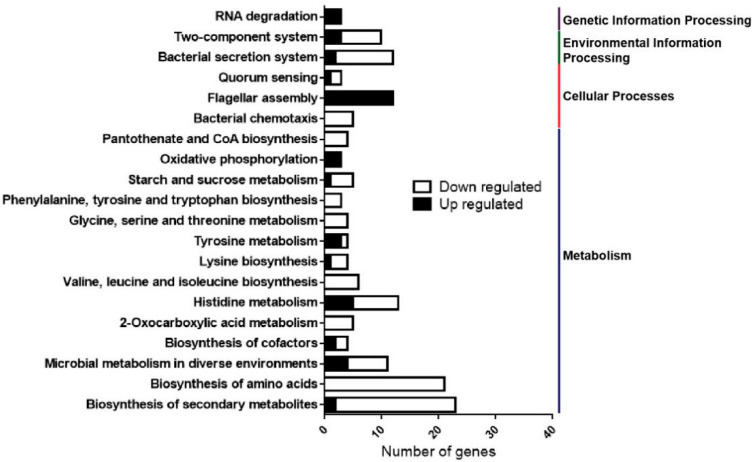
The Kyoto Encyclopedia of Genes and Genomes (KEGG) enrichment analysis of pathways regulated by HemK. The number of annotated genes (*x*-axis) versus KEGG categories (*y*-axis).

**Figure 6 ijms-23-03931-f006:**
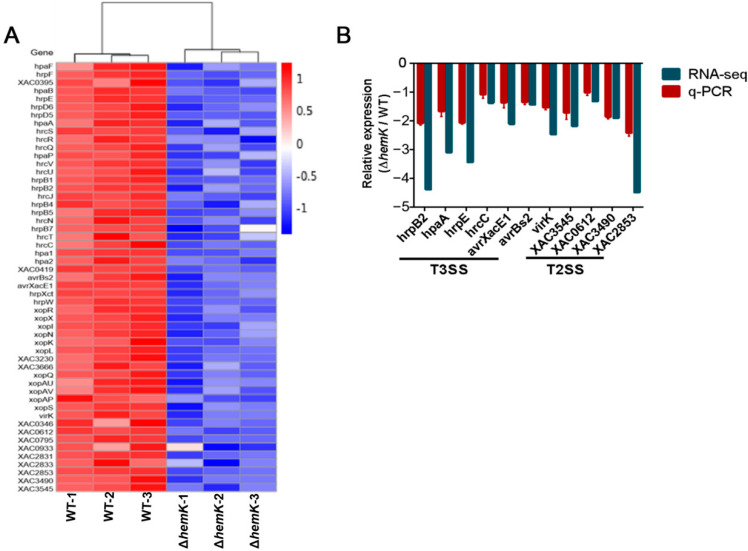
HemK regulates the expression of T3SS- and T2SS-associated genes. (**A**) Heat map of gene expression of T2SS and T3SS by RNA-seq data. (**B**) The relative expression of genes associated with the virulence identified by RNA-seq analysis was determined by qRT-PCR analysis. The target genes included *hrpB2*, *hpaA*, *hrcC* and *hrpE* (T3SS regulators), *virK*, *avrXacE1* and *avrBs2* (virulence protein), *XAC3545* (protease), *XAC0612* (cellulase), *XAC3490* (amylosucrase or alpha-amylase) and *XAC2853* (cysteine protease).

**Table 1 ijms-23-03931-t001:** List of genes related to quorum sensing in *X. citri* subsp. *citri* jx-6, regulated by HemK.

Locus Tag	Gene Name	Log_2_Fold Change	Protein Function
XAC2992	XAC2992	−1.5	endoproteinase Arg-C, degrading host defense proteins
XAC0612	engXCA	−1.3	cellulase
XAC3120	glk	1.3	glucose kinase
XAC3921	ugt	−1.2	glucosyltransferase
XAC1556	fucP	−1.3	glucose-galactose transporter
XAC1557	scrK	−1.2	fructokinase
XAC3489	fyuA	−2.3	TonB-dependent receptor
XAC3490	XAC3490	−1.9	amylosucrase or alpha amylase
XAC0465	XAC0465	−1.0	metalloproteinase
XAC4327	uahA	−1.8	urea amidolyase
XAC1820	metL	−3.3	aspartokinase
XAC1821	thrB	−3.4	homoserine kinase
XAC1823	thrC	−3.6	threonine synthase
XAC1828	hisG	−2.4	ATP phosphoribosyltransferase
XAC1829	hisD	−2.3	histidinol dehydrogenase
XAC1830	hisC	−2.4	histidinol-phosphate aminotransferase
XAC1831	hisB	−2.5	Imidazole glycerol phosphate dehydratase/histidinol-phosphate phosphatase bifunctional enzyme
XAC1832	hisH	−1.9	amidotransferase
XAC1833	hisA	−2.2	phosphoribosylformimino-5-aminoimidazole carboxam
XAC1834	hisF	−2.3	cyclase
XAC1835	hisI	−2.4	phosphoribosyl-AMP cyclohydrolase/phosphoribosyl—ATP pyrophosphatase bifunctional enzyme
XAC3451	ilvC	−2.7	ketol-acid reductoisomerase
XAC3452	ilvG	−2.2	acetolactate synthase isozyme II large subunit
XAC3453	ilvM	−2.5	acetolactate synthase isozyme II large subunit
XAC3454	tdcB	−2.1	threonine dehydratase catabolic
XAC3455	leuA	−1.3	2-isopropylmalate synthase
XAC0999	cirA	−1.1	colicin I receptor
XAC3546	xadA	−1.7	autotransporter adhesion protein
XAC1471	XAC1471	−1.1	glycine zipper 2TM domain containing protein
XAC1827	XAC1827	−3.2	Trp repressor protein
XAC3085	XAC3085	−1.1	putative type III secretion system effector protein
XAC3754	XAC3754	1.4	putative chemotaxis membrane protein

## Data Availability

Not applicable.

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
