# Peer review of "The Methyltransferase HemK Regulates the Virulence and Nutrient Utilization of the Phytopathogenic Bacterium Xanthomonas citri Subsp. citri"

_ijms, 2022, doi:10.3390/ijms23073931_

Round 1

Reviewer 1 Report

This paper studies HemK and demonstrates that it is important for pathogen virulence. Deleting hemK decreased pathogen motility, biofilm formation and increases its sensitivity to certain stresses besides other things. Comparative transcriptomics showed that hemK deletion changed the expression of several genes, more importantly the ones related to the type III secretion system. However, there some concerns that must be attended before the manuscript can be accepted for publication.

Figures are of poor resolution and would pixelate on slight enlargement. For example in Fig1B, I could not see what is written in red on the plates.

Line 105-106 This may be used instead “Mutant ΔhemK showed considerable reduction in both swimming and swarming motility compared with wild-type strain”.

Line 106-107 correct this sentence

Line 109 quantify = quantifying

Line 117 significant= significantly

Line 287 Please mention in the preceding para that T2SS genes are differentially expressed (just like you have done for T3SS) so that a connection is made.

Line 287-288 ‘To further confirm whether HemK is required for regulation of T3SS- and T2SS-associated genes expression revealed by the RNA-seq analysis, we conducted quantitative real-time PCR (qRT-PCR)…..’. Which of these genes is T2SS genes, that you have alluded to? You may Cite Figure 6A (heatmap) there if it contains T2SS genes.

In Fig6B qRT-PCR, there is no WT to show relative expression. It would help the readers to put WT in addition to the RNA-seq data.

Line 303-307 It would make more sense to move these lines ahead to give a better understanding of what follows next.

I think the results section 2.6 does not add any value to the manuscript and can be removed (the information in supplementary data seems enough) and then section 2.7 can be merged with section 2.5.

Author Response

Comments and Suggestions for Authors

This paper studies HemK and demonstrates that it is important for pathogen virulence. Deleting hemK decreased pathogen motility, biofilm formation and increases its sensitivity to certain stresses besides other things. Comparative transcriptomics showed that hemK deletion changed the expression of several genes, more importantly the ones related to the type III secretion system. However, there some concerns that must be attended before the manuscript can be accepted for publication.

  1. Figures are of poor resolution and would pixelate on slight enlargement. For example in Fig1B, I could not see what is written in red on the plates.

Authors response: We are highly thankful to the reviewer for the advice. As suggested, we changed the color of the words in Fig1B from red to white and made them bigger.

  1. Line 105-106 This may be used instead “Mutant ΔhemK showed considerable reduction in both swimming and swarming motility compared with wild-type strain”.

Authors response: We are highly thankful to the reviewer for the modification. We have revised the sentence according to the comment.

  1. Line 106-107 correct this sentence

Authors response: We are highly thankful to the reviewer for the advice. We have corrected the sentence as follows: In the complementary strain C-∆ hemK, these reduced motilities can be restored.

  1. Line 109 quantify = quantifying, Line 117 significant= significantly,

Authors response: We are highly thankful to the reviewer’s correction. We have corrected the words in the manuscript.

  1. Line 287 Please mention in the preceding para that T2SS genes are differentially expressed (just like you have done for T3SS) so that a connection is made.

Authors response: We are highly thankful to the reviewer for the advice. Now we put the T2SS genes in the preceding paragraph which includes “RNA-seq data also revealed in ∆hemK a decreased expression of 10 genes encoding bacterial exoenzymes secreted through T2SS (virK, XAC0346, XAC0612, XAC0795, XAC0933, XAC2831, XAC2833, XAC2853, XAC3490 and XAC3545) (Figure 6A, Supplementary Table S1).”.

  1. Line 287-288 ‘To further confirm whether HemK is required for regulation of T3SS- and T2SS-associated genes expression revealed by the RNA-seq analysis, we conducted quantitative real-time PCR (qRT-PCR)…..’. Which of these genes is T2SS genes, that you have alluded to? You may Cite Figure 6A (heatmap) there if it contains T2SS genes.

Authors response: We are highly thankful to the reviewer for the advice. We have list the T2SS genes as a new paragraph after the T3SS genes. Which is “RNA-seq data also revealed in ∆hemK a decreased expression of 10 genes encoding bacterial exoenzymes secreted through T2SS (virK, XAC0346, XAC0612, XAC0795, XAC0933, XAC2831, XAC2833, XAC2853, XAC3490 and XAC3545) (Figure 6A and Supplementary Table S1).”

  1. In Fig6B qRT-PCR, there is no WT to show relative expression. It would help the readers to put WT in addition to the RNA-seq data.

Authors response: We are highly thankful to the reviewer’ advice. We changed the description of the Fig6B y-axis by adding (∆hemK /WT) after the relative expression, which means the gene expression level of ∆hemK mutant compared with that of wild-type.

  1. Line 303-307 It would make more sense to move these lines ahead to give a better understanding of what follows next.

Authors response: We are highly thankful to the reviewer’ advice. As suggested, we moved the lines forward and separated the T2SS genes as a new paragraph, which gives a better understanding of what follows next.

  1. I think the results section 2.6 does not add any value to the manuscript and can be removed (the information in supplementary data seems enough) and then section 2.7 can be merged with section 2.5.

Authors response: We are highly thankful to the reviewer for the nice suggestion. As suggested, we removed the result section 2.6 and merged some data with section 2.5.

Reviewer 2 Report

In this investigation, Authors stated that the functional involvement of HemK, a glutamine methyltransferase from Xcc in virulence and its regulation of gene expression during disease processes. Mutant ΔhemK displayed several virulence-related phenotypes manifested by reduced motility, extracellular enzymes and polysaccharides production, biofilm formation and pathogenicity. This discovery provides new information on the pathogenicity of this important plant pathogen. The manuscript is well structured and well discussed. However, some points should be checked and corrected before its acceptance in this journal. 

Therefore, I recommended the publications of the paper after major revision according to given my comments.

  • The study's background should be clearly stated. Describe the introduction and review of the work.
  • Please revise figure 4. Please explain what is this number 12, 17, 19 etc.?
  • Please speculate on the results. The discussion must improve.
  • In Conclusion, the authors should add the significance of this research, and its potential practical application.
  • The MS English needs to be improved. The article's English must be carefully checked for grammatical errors.

Author Response

In this investigation, Authors stated that the functional involvement of HemK, a glutamine methyltransferase from Xcc in virulence and its regulation of gene expression during disease processes. Mutant ΔhemK displayed several virulence-related phenotypes manifested by reduced motility, extracellular enzymes and polysaccharides production, biofilm formation and pathogenicity. This discovery provides new information on the pathogenicity of this important plant pathogen. The manuscript is well structured and well discussed. However, some points should be checked and corrected before its acceptance in this journal. 

Therefore, I recommended the publications of the paper after major revision according to given my comments.

Authors response: Thank you very much for your valuable comments and for offering us the opportunity to improve the manuscript for potential publication. We have followed all the suggestions and revised the manuscript accordingly.

  1. The study's background should be clearly stated. Describe the introduction and review of the work.

Authors response: We are highly thankful to the reviewer for the advice. As suggested, we have added more study background in the introduction section.

  1. Please revise figure 4. Please explain what is this number 12, 17, 19 etc.?

Authors response: We are highly thankful to the reviewer for the advice. We have revised the Figure 4 legend and described how many differentially expressed genes (DEGs) the numbers (12, 17, 19, etc.) mean between the Δhemk and Xcc jx-6 strains.

  1. Please speculate on the results. The discussion must improve.

Authors response: We are highly thankful to the reviewer for the advice. As suggested, we made some speculations in the Results section and revised the Discussion section.

  1. In Conclusion, the authors should add the significance of this research, and its potential practical application.

Authors response: We are highly thankful to the reviewer for the advice. As suggested, we have added the significance of this study and its potential practical applications in the Introduction and Discussion sections.

  1. The MS English needs to be improved. The article's English must be carefully checked for grammatical errors.

Authors response: We are highly thankful to the reviewer for the advice. We have revised the manuscript and carefully proofread it to minimize typographical, grammatical, and bibliographic errors.

Round 2

Reviewer 2 Report

Requested corrections were completed.